# The Role of microRNA-22 in Metabolism

**DOI:** 10.3390/ijms26020782

**Published:** 2025-01-17

**Authors:** Simone Tomasini, Paolo Vigo, Francesco Margiotta, Ulrik Søberg Scheele, Riccardo Panella, Sakari Kauppinen

**Affiliations:** 1Center for RNA Medicine, Department of Clinical Medicine, Aalborg University, 2450 Copenhagen, Denmark; sito@dcm.aau.dk (S.T.); ulrik.scheele@gmail.com (U.S.S.); riccardop@dcm.aau.dk (R.P.); 2Resalis Therapeutics Srl, Via E. De Sonnaz 19, 10121 Torino, Italy; 3Pharmacology and Toxicology Section, Department of Neuroscience, Psychology, Drug Research and Child Health (NEUROFARBA), University of Florence, Viale G. Pieraccini 6, 50139 Florence, Italy; francesco.margiotta@unifi.it; 4European Biomedical Research Institute of Salerno (EBRIS), Via Salvatore de Renzi 50, 84125 Salerno, Italy

**Keywords:** microRNA, miRNA, miR-22, therapy, therapeutics, RNA medicine, metabolism, obesity, DMD, Duchenne muscular dystrophy, cardiac health, MASLD, MASH, T2D

## Abstract

microRNA-22 (miR-22) plays a pivotal role in the regulation of metabolic processes and has emerged as a therapeutic target in metabolic disorders, including obesity, type 2 diabetes, and metabolic-associated liver diseases. While miR-22 exhibits context-dependent effects, promoting or inhibiting metabolic pathways depending on tissue and condition, current research highlights its therapeutic potential, particularly through inhibition strategies using chemically modified antisense oligonucleotides. This review examines the dual regulatory functions of miR-22 across key metabolic pathways, offering perspectives on its integration into next-generation diagnostic and therapeutic approaches while acknowledging the complexities of its roles in metabolic homeostasis.

## 1. Introduction

MicroRNAs (miRNAs) are an abundant class of small (~22 nt) endogenous non-coding RNAs that act as important post-transcriptional regulators of gene expression. Mature miRNAs mediate gene silencing by directing the RNA-Induced Silencing Complex (RISC) to target transcripts leading to translational repression and/or mRNA degradation [1]. miRNAs are frequently dysregulated in diseases such as cancer, central nervous system (CNS) disorders, inflammation, and cardiometabolic diseases [2,3,4,5,6]. As a result, miRNAs have emerged as promising diagnostic biomarkers and molecular targets for RNA-based therapeutics [7]. The therapeutic potential is particularly enhanced by the ability to exert the silencing function beyond the native cell, since miRNAs are actively secreted in the extracellular space within extracellular vesicles (EVs) [7,8]. miRNAs can thus mediate both autocrine and paracrine signaling within single tissues, but also hold an endocrine role in distal organs, when EVs reach the bloodstream. Pharmacologically, the impact of miRNA biology has been investigated in the treatment of several diseases as it might establish a broader and longer lasting therapeutic effect [9]. On the other hand, circulating miRNAs also show promise as readily detectable biomarkers in blood and other body fluids, enabling less invasive and more sensitive detection for early and more precise diagnosis of a wide variety of diseases [10,11,12].

MicroRNA-22 (miR-22) is a highly conserved and ubiquitously expressed miRNA, present from *Drosophila melanogaster* to humans, originating from exon 2 of the non-coding host gene *MIR22HG*. During the maturation of miR-22, both strands of the duplex (miR-22-5p and miR-22-3p) are generated. However, miR-22-3p predominantly serves as the functional guide strand, targeting specific mRNAs, while miR-22-5p is typically the passenger strand and subject to degradation. The predominance of miR-22-3p is attributed to its higher stability, enriched expression in metabolically active tissues, and stronger association with regulatory pathways, as evidenced in studies demonstrating its role in suppressing tumorigenesis and regulating metabolic processes. Conversely, miR-22-5p shows limited functional relevance in metabolism-related contexts, although it has been implicated in certain cardiovascular and oncological pathways [13,14]. In this review, we will focus on miR-22-3p, which will be referred to as miR-22 in the rest of the manuscript.

The *MIR22HG* gene is located at 17p13.3, a chromosomal region frequently deleted or transcriptionally silenced in cancer [15,16]. It encodes four long non-coding RNAs (lncRNAs), which have been investigated for their inhibitory effect on cell proliferation, migration, and invasion in different types of cancer by negatively regulating Wnt/β-catenin, Notch, STAT3, and TGFβ signaling pathways [17,18,19,20]. Further complexity in the regulation of *MIR22HG* antitumoral activity derives from its ability to sequester multiple miRNAs with oncogenic potential, such as miR-9-3p [21], miR-10-5p, and miR-24-3p [22,23,24,25], and to regulate miR-22 expression. In turn, miR-22 is involved in several aspects of cancer biology, being either up- or downregulated depending on the malignancy and, consequently, acting either as a tumor suppressor or as an oncogene. miR-22 is mainly known to control cancer formation and progression by influencing cell growth, epithelial-to-mesenchymal transition, senescence, and inflammation [26,27,28].

This review will focus on miR-22 as a metabolic regulator in relevant tissues, and on its regulatory function in key pathways such as gluconeogenesis, hepatic steatosis, mitochondrial biogenesis and fatty acid biosynthesis, often dysregulated in metabolic syndrome. We will review recent studies implicating miR-22 in the pathogenesis of the most important metabolic diseases such as obesity, type 2 diabetes (T2D), metabolic dysfunction-associated steatotic liver disease (MASLD), and steatohepatitis (MASH). Furthermore, we will discuss its importance in muscle homeostasis, focusing on cardiac development and regeneration to introduce the potential of miR-22 as a biomarker and a therapeutic target for a wider group of metabolic conditions. Notably, miR-22 exhibits a context-dependent regulatory role, influencing metabolic and muscular pathways in distinct and tissue-specific manners. Understanding how miR-22 integrates these signals is critical to fully elucidating its therapeutic potential.

## 2. miR-22 as a Regulator of Key Metabolic Factors

Given the multifaceted roles of miRNAs in cellular regulation, miR-22 stands out as a central regulator within metabolic pathways, particularly in tissues essential for energy homeostasis, such as the liver, adipose tissue, and skeletal muscle (Figure 1). Its enrichment in metabolically active tissues, including the kidney and cardiac muscle, supports its critical role in coordinating energy storage and expenditure, underscoring its potential as a therapeutic target for metabolic disorders [29,30,31,32,33,34,35]. Gene ontology (GO) analysis of predicted targets of miR-22 conserved in humans and mice uncovers a strong metabolic and regenerative component with high enrichment in the central carbon metabolism pathway [36]. Many genes post-transcriptionally regulated by miR-22 are key players in glucose metabolism, lipogenesis, thermogenesis, and mitochondrial homeostasis [29,37,38,39,40,41,42,43,44]. For instance, miR-22 modulates the expression of peroxisome proliferator γ-activated receptor coactivator 1-α (PGC-1α), which is important for hepatic glucose production (HGP) in the liver by regulating gluconeogenesis-related genes such as PEPCK and G6Pase [45,46]. Since the acetylation and phosphorylation of PGC-1α result in its partial inhibition, the removal of post-translational modification improves HGP [47]. miR-22 also targets NAD-dependent protein deacetylase sirtuin-1 (SIRT1), which controls gluconeogenesis by derepressing PGC-1α and inhibiting the signal transducer and activator of transcription 3 (STAT3) [48,49]. miR-22 contributes to lipogenesis by controlling the expression of crucial enzymes involved in fatty acid synthesis like fatty acyl-CoA elongase (ELOVL6) and fatty acid synthase (FASN) [50]. It also regulates mitochondrial fatty acid β-oxidation by targeting peroxisome proliferator-activated receptor alpha (PPAR-α) and its associated enzymes [51,52]. PPAR-α controls the expression of several enzymes involved in both peroxisomal and mitochondrial β oxidation like ACOX1, EHHADH and MCAD, LCAD, and CPT1A, respectively [53,54,55]. PGC-1α, Lipin 1, and SIRT-1 are all coactivators of PPAR-α and positively modulate its transcriptional activity in the liver by binding the protein and promote FAO during the fasted state [56,57]. Prolonged fasting and PPARα activation also induce the secretion of the hepatokine fibroblast growth factor 21 (FGF21), to further stimulate FAO and ketogenic programs by boosting PGC-1α transcription in the liver [58]. The release of FGF21 also affects the expression of *PPARGC1A* (PGC-1α-encoding gene) in adipose tissue (AT), where it mediates the activation of non-shivering thermogenic pathways and browning of white adipose tissue (WAT). During such adaptive processes, mature white adipocytes progressively commit to a high energy expenditure program, increasing the synthesis of brown adipose tissue (BAT) markers, such as elevated mitochondrial biogenesis and multilocularization of lipid droplets [59,60]. miR-22 has been implicated in the regulation of adipogenesis in WAT, by reducing the expression of key enzymes responsible for fatty acid synthesis and adipocyte differentiation [29] and driving the interconversion to beige adipocytes [36,40,41]. Possible mechanisms in the control of brownization programs rely on the modulation of PGC-1α, which is a well-characterized regulator of mitochondrial biogenesis and thermogenesis [61,62,63]. The activation of PGC-1α by SIRT1 has been reported to induce mitochondrial biogenesis via protein–protein interaction with nuclear respiratory factors (NRF1 and NRF2), which are upregulated upon miR-22 inhibition in murine myoblasts [64]. The resulting transcriptional coactivator complex promotes the expression of genes involved in the replication, transcription, and maintenance of mitochondrial DNA (mtDNA), such as mitochondrial transcription factor A (*TFAM*) [65,66,67]. Furthermore, PGC-1α is responsible for the activation of uncoupling protein 1 (UCP1), a key factor in non-shivering thermogenesis, by the joined transactivation of its promoter region with IRF4 [68,69]. UCP1 is responsible for mitochondrial uncoupling in BAT, redirecting the use of the mitochondrial proton gradient from ATP synthesis through oxidative phosphorylation (OXPHOS) to heat generation and energy dissipation. Thus, the transcript and protein levels of UCP1 can be used as biomarkers for BAT and browning, the latter of which in conjunction with an elevated expression of *PPARGC1A*, zinc finger protein 516 (*ZNF516*), and the PR domain containing 16 genes (*PRDM16*) [70]. Both *TFAM* and *UCP1* are upregulated in BAT compared to WAT and their increased expression is sufficient to drive brownization, whereas their absence promotes an opposite whitening processes [71,72,73] (see [40]). Interestingly, transgenic mice specifically lacking miR-22 in either AT or BAT revealed an opposite cellular response in BAT [74]. Brown adipocytes were subjected to extensive unilocular lipid accumulation and decreased glycolytic function, which was rescued by the targeted overexpression of miR-22 in these tissues [74]. Notably, the expression of miR-22 target genes such as *Ppargc1a* in the aforementioned mice followed an opposite trend than what has been reported in the literature [37,52,75,76], showing the upregulation of *Ppargc1a* upon miR-22 overexpression and the opposite effect in ablated animals. This suggests that metabolic rewiring towards higher energy expenditure may depend on other metabolically active tissues such as the liver and skeletal muscles.

miR-22 is implicated in the control of lipid metabolism in skeletal muscle, which is an important player in basal energy expenditure and a metabolically active tissue accounting for almost half of the body mass. Schweisgut et al. identified miR-22 as a possible driver of the sex-specific regulation of muscular lipid metabolism and body weight by modulating the expression of estrogen receptor alpha (ERα), a known promoter of lipid catabolism. Genetic ablation of miR-22 in mice induced a reduction in WAT and body weight, specifically in males, by abolishing the negative control over ERα expression and increasing fatty acid oxidation in the muscle. Importantly, the absence of miR-22 does not affect muscle mass, but only the fat cell dimension [77]. Further evidence on the importance of miR-22 controlling oxidative programs in the muscle tissue derives from in vitro studies associating this miRNA to muscle fiber-type conversion. The clinical relevance of fiber-type switching relates to a higher resilience of oxidative fibers to chronic injury, a hallmark in the pathogenetic progression of muscular dystrophies and myopathies [78,79]. miR-22 regulates muscle fiber composition by modulating the AMPK/SIRT1/PGC-1α axis. High levels of miR-22 promote the enrichment of fast-twitch, type II glycolytic myofibers, which are associated with reduced oxidative capacity. Conversely, the inhibition of miR-22 leads to an increase in slow-twitch oxidative fibers (type I), characterized by enhanced mitochondrial biogenesis and improved aerobic capacity. Improvements in muscle aerobic capacity have been shown to be dependent on SIRT1/PGC-1α activation, with an increased expression of type I fiber-specific genes and mitochondrial tissue density upon treatment with PGC-1α activators such as resveratrol [80,81]. miR-22 is also an activator of the rapamycin (mTOR) signaling pathway, a major nutrient sensor, which promotes the switch from catabolism to anabolism following the increased energy and endocrine signaling mainly induced by feeding [82]. miR-22 regulation derives from the direct inhibition of two negative modulators of this pathway, PTEN and REDD1 [83,84], that act upstream and downstream of the serine-threonine protein kinase AKT, respectively [85]. Skeletal muscle is the principal muscle responsible for glucose homeostasis controlling glucose uptake via insulin-dependent and -independent mechanisms. Human type I muscle fibers have a higher glucose-handling capacity compared to type II fibers [86]. Basal glucose uptake by the muscle has been demonstrated to be proportional to the expression of the glucose transporters GLUT4 and GLUT1 in mice, where GLUT1 overexpression leads to a substantial decrease in fasting blood glucose levels and increase in glucose tolerance [87,88,89]. By controlling GLUT1 expression, miR-22 could function as a putative regulatory element of glucose metabolism and insulin sensitivity, as gain-of-function mutation in GLUT1 promotes the activation of different members of the insulin pathway [90,91]. Furthermore, circadian disruption stress was recently observed to be causative of higher levels of circulating miR-22 and IR-relative factors due to increased adipogenesis in mice and humans. Zhang et al. have proposed a model, where miR-22 controls the crosstalk between adipose tissue and skeletal muscle to regulate insulin-mediated glucose uptake in mice. In accordance with this model, the endocytosis of adipocyte-derived exosomal miR-22 hinders glucose internalization in myotubes by partially inactivating the insulin signaling pathway [92]. Although the mechanism is plausible depending on the decreased expression of *Glut1*, further work is required to demonstrate the role of miR-22 in the physiological response to postprandial blood glucose increase.

Various treatments have been shown to modulate miR-22 levels and their regulatory effects on metabolic pathways (Table 1). Antagonism of miR-22 was shown to repress pathologic increased levels of PEPCK and G6Pase as well as fasting glucose levels in mice via the Tcf7/β-catenin axis, possibly through forkhead box protein O1 (FOXO1) [38]. Furthermore, modulation of miR-22 was also suggested to be involved in the mechanism of action of 3,5-diiodo-L-thyronine (T2), an active iodothyronine implicated in glucose homeostasis. Administration of T2 activates hepatic SIRT1 directly and causes a decrease in miR-22 levels in both the serum and liver of rats fed on a diet with high fat content, impairing gluconeogenesis possibly through the derepression of *TCF7* [48,49]. In addition to glucose regulation, targeting miR-22 influences lipid metabolism and thermogenesis. The genetic ablation and pharmacologic inhibition of miR-22, respectively, result in the profound activation of browning of visceral and subcutaneous fat, as well as the enhanced expression of thermogenic genes in BAT, especially when the mice are challenged with a high-fat diet [39]. Further evidence supports the role of miR-22 antagonism in promoting an oxidative metabolic program. Greene et al. demonstrated that antagonism of miR-22 promotes an oxidative metabolic program in mice, consistent with SIRT1/PGC-1α activation. Similarly, in C2C12 cells, miR-22 overexpression partially counteracts this oxidative shift, highlighting the dual regulatory role of miR-22 in muscle metabolism [84,85,86,87,88]. Interestingly, miR-22 is also involved in the regulation of muscle fiber composition through the mTOR pathway. The constitutive induction of the mTOR pathway was also shown to induce aerobic glycolysis and conversion to fast-twitch, type II fibers in murine skeletal muscles, similar to the effect induced by miR-22 overexpression [93].

These findings collectively highlight the therapeutic potential of targeting miR-22 to modulate its effects in various metabolic contexts.

## 3. Role of miR-22 in Diabetes and Obesity

Building on the understanding of the regulatory role of miR-22 in metabolic pathways, its impact extends to specific metabolic disorders such as diabetes and obesity, which are major risk factors for cardiovascular disease—the leading cause of mortality worldwide [93,100]. The high prevalence and well-documented association of these conditions with miRNA dysregulation underscore the potential significance of miR-22 in disease pathogenesis and therapeutic intervention.

Over the past few decades, the prevalence of the abovementioned conditions has gradually increased, posing a significant burden to public health [101]. Metabolic syndrome is diagnosed as a co-occurrence of several metabolic factors including elevated adiposity and blood pressure, impaired fasting blood glucose, and abnormal high-density lipoprotein cholesterol and triglyceride levels. It increases the risk of developing cardiovascular disease, cerebrovascular accidents, and T2D [102], which represents a major socioeconomic burden among all chronic metabolic conditions. T2D is characterized by hyperglycemia and dyslipidemia resulting from the progressive development of chronic insulin resistance and insulin deficiency. T2D affects more than 480 million adults globally, a number that is estimated to rise by 2045 especially as an early-onset form, with higher tolls likely to be paid by low- and middle-income countries, which lack the infrastructure for prevention and early diagnosis [103]. T2D is a heterogeneous disorder caused by the interplay between genetic factors and excessive nutrient consumption. Comorbidities such as hypertension, hyperlipidemia, or obesity—whether individually or in combination—are observed in the majority of T2D patients at diagnosis [104,105], highlighting the importance of increased lipid content in the disease onset. Indeed, sustained weight loss of 5–10% can delay the progression from prediabetes to diabetes [103]. Obesity is the result of excessive fat accumulation in the body. Diagnosis still relies on Body Mass Index (BMI) and waist circumference cut-offs, although more precise methodologies to assess body fat adiposity are present and recommended. Like T2D, obesity has seen a dramatic global increase, with its prevalence more than tripling over the past 40 years [106,107].

miR-22 has been shown to be upregulated in the liver and adipose tissue of mouse models of obesity and T2D [39,108,109]. Indeed, as we previously demonstrated, miR-22 overexpression is sufficient to induce obesity in mice, leading to an even more extreme phenotype when its overexpression is specifically confined to hepatocytes [36]. On the other hand, genetic deletion (KO) of exon 2 of *MIR22HG* is not sufficient per se to prevent body weight gain in mice fed on a high-saturated-fat diet (HFD), but it reduces WAT expansion and cellular triglyceride (TG) content compared to wildtype (WT) littermates. This effect is only present in obese mice and no difference in either adipocyte size nor TG is observed between WT and miR-22 KO mice fed on normal chow [40]. Interestingly, the pharmacological inhibition of miR-22 in diet-induced obesity (DIO) mouse models significantly slows weight gain mediated by HFD, suggesting that the promotion of energy expenditure programs driven by miR-22 inhibition is dependent on high energy intake [29,36,42]. Furthermore, inhibition of miR-22 alleviates random and fasting hyperglycemia, improving glucose tolerance and insulin sensitivity both in WT DIO mice and the db/db T2D mouse model [18,42].

The role of miR-22 in controlling adipogenesis was linked to its indirect control over key transcription factors and enzymes involved in de novo lipogenesis (DNL), such as SREBP-1c, PPARγ, and FASN [29,37,40]. The increased hepatic expression of these genes is clinically relevant in obesity [110,111], and their genetic inhibition has a protective effect on liver steatosis [112,113]. SREBP-1c expression was shown to positively correlate with miR-22 levels in cultured human hepatocytes, resulting in increased expression under treatment with free fatty acids (FFAs) [43]. Conversely, the pharmacological inhibition of miR-22 restores SREBP-1c expression to pre-obesogenic levels and possibly further inhibits SREBP-1c activity through direct transcriptional derepression of SIRT1 [114], further supporting miR-22 involvement in lipid metabolism regulation [43]. The transcriptional regulation of SREBP-1c by miR-22 was also validated in a genetic mouse model of total-body miR-22 KO [29]. Despite this mechanism still having to be elucidated, it can be speculated that *SREBP-1* expression could be controlled by a genetic target of miR-22. Indeed, *SREBP-1* is downregulated by several nuclear receptors, among which is ERα [99], a direct target of miR-22. Studies are needed to confirm the possible role of miR-22 in driving steatosis via ERα and SREBP-1c modulation. Furthermore, genetic ablation of miR-22 transcriptionally inhibits *Pparg*, another master regulator of lipogenesis in the liver, therefore acting to reduce DNL in a multi-pathway manner [29].

In the adipose tissue, PPARγ stimulates adipocyte differentiation [115], and its deletion was shown to protect against HFD, limiting the expansion of WAT [116]. miR-22 expression was observed to increase during the maturation of adipocytes; in vitro genetic ablation of miR-22 in pre-adipocytes causes the reduced expression of different adipogenic markers, among which is *Pparg*, impairing terminal differentiation [40]. The same modifications in the transcriptional profile of adipocytes were validated in miR-22 KO mice fed HFD, showing the reversion of the increased cell size in subcutaneous and visceral WAT, as well as triglyceride accumulation due to the diet, to match WT littermates on normal diet [29,40].

These findings suggest that miR-22 may influence adipogenesis through different tissue-specific pathways controlled by key transcription factors. miR-22 modulation in response to HFD correlates with the concurrent silencing of essential components in lipid metabolism and adipogenesis, emphasizing its role in the metabolic dysregulation observed under obesogenic conditions. Together, these data underline the importance of miR-22 as a regulator of SREBP-1c and other critical factors involved in lipid biosynthesis and energy homeostasis.

Several direct miR-22 targets are well-characterized factors in different aspects of metabolic homeostasis and their decreased expression has been linked to the development and progression of obesity, T2D, and MS. Notably, PGC-1α and SIRT1—key regulators of gluconeogenesis, lipogenesis, and thermogenesis, respectively—are downregulated in the adipose tissue of individuals with obesity and T2D [43,117,118]. Deletion of miR-22 was shown to restore and increase the expression of *PPARGC1A*, together with other BAT markers such as *PRDM16*, *CIDEA*, and *ELOVL6*, in the WAT of mice fed HFD. Moreover, miR-22 KO eliminated the diet-induced alteration observed in the oxygen consumption rates of mitochondria isolated from the BAT of miR-22 KO animals, implying that the absence of miR-22 contributes to reverting the obesogenic condition by promoting higher energy expenditure [40]. In the liver, miR-22 has an opposite effect on mitochondria biogenesis compared to its role in adipose tissue, counteracting the effect of a western diet. While HFD increases mitochondrial content in the liver of wild-type mice, as indicated by the mtDNA-to-genomic DNA (gDNA) ratio, this effect is abolished with miR-22 genetic deletion. miR-22-KO mice show mtDNA/gDNA ratios comparable to those of WT mice on a control diet [29]. This observation suggests that the therapeutic inhibition of miR-22 can protect against obesity-driven mitochondrial dysfunction.

A recent study on glucose and lipid metabolism exploiting farnesoid X receptor agonists [119] highlights the mechanism of gluconeogenesis suppression by the miR-22/PI3K/AKT/FOXO1 pathway in T2DM mice, through a decrease in miR-22 expression and an increase in FOXO1 levels. In addition, miR-22 has an impact on glycogen synthesis through the miR-22/PI3K/AKT/GSK3β pathway [119].

The role of miR-22 in diabetes is further supported by the role of miR-22 in impairing gluconeogenesis, through the downregulation of TCF7 [39]. Moreover, miR-22 inhibition has been demonstrated to lower random and fasting glucose levels in diabetic mice, improving glucose and insulin tolerance in vivo [39]. The same results were observed in a rat model of diabetes, in which miR-22 levels inversely correlate with TCF7 expression and promote the transcriptional inhibition of enzymes involved in gluconeogenesis [96]. Taken together, these findings suggest that the inhibition of miR-22 function can reduce fat mass without affecting food intake or body temperature (see [41]). Importantly, there is no evidence linking this effect to the accumulation of miR-22 in the bloodstream as a consequence of liver damage. Instead, it reflects the role of miR-22 in the regulation of adipogenesis and energy homeostasis.

## 4. Role of miR-22 in MASLD and MASH

In addition to its roles in systemic metabolism, miR-22 has significant implications in liver health, particularly concerning metabolic dysfunction-associated steatotic liver disease and metabolic-associated steatohepatitis. These liver diseases are critical areas of study due to their close association with metabolic syndrome and their progression towards severe liver pathology, highlighting miR-22 as both a biomarker and therapeutic target in this context.

The accumulation of ectopic fat is a major risk factor for the development of hepatic steatosis, being likely to be diagnosed as a BMI increase [120]. As a result, non-alcoholic fatty liver disease (NAFLD) has been the most common chronic liver disease in the western world unrelated to alcohol consumption, with an estimated prevalence of 25% [121,122]. NAFLD is often accompanying metabolic risk factors like obesity and type 2 diabetes [123,124], and is characterized by progressive hepatic degeneration. Stage one is diagnosed as hepatic steatosis in over 5% of hepatocytes in the absence of a daily alcohol intake exceeding 30 g for men and 20 g for women and other known causes of chronic liver disease [125,126]. Lately, the nomenclature has been changed to metabolic dysfunction–associated fatty liver disease (MAFLD), and more recently to MASLD, as steatotic liver disease (SLD) is the new umbrella term to include a larger group of morbidities sharing similar etiology regardless of the pathology and avoiding stigmatic language [127]. Moreover, MASLD better reflects the importance of the interplay between hepatic steatosis, cardiovascular disease, and T2D, as MASLD diagnosis is restricted to NAFLD patients affected by at least one cardiometabolic risk factor [128]. Similarly, non-alcoholic steatohepatitis (NASH), which represented the advanced stage of NAFLD, has been replaced by MASH, which is characterized by increased hepatic steatosis, ballooning, and the presence of persistent lobular inflammation. Almost a third of people diagnosed with MASLD are estimated to progress to MASH, which is still a reversible condition [129]. A lack of intervention results in increased hepatic fibrosis and scarring eventually leads to the latest stages of the disease, encompassing hepatic failure and the development of hepatocellular carcinoma. Effective pharmacologic intervention to halt and reverse the pathophysiology of MASLD and MASH is therefore needed to avoid liver transplantation in affected patients. As most of the literature regarding the role of miR-22 in SLD was generated under the NAFLD/NASH era, this article will focus on SLD caused by external factors other than hepatitis virus infection, autoimmune diseases, and alcohol abuse.

miR-22 is highly expressed in subjects with MASLD [29,35,37]. Many studies point towards miR-22 promoting hepatic steatosis, and pharmacologic intervention against miR-22 counteracts SLD acting on different pathways [43]. Furthermore, the deletion of miR-22 was shown to be beneficial in reverting the steatogenic effect of HFD in mice, decreasing liver weight and hepatic fat accumulation, suggesting protection against liver steatosis [74].

miR-22 can regulate the inflammatory effects of steatosis by increasing the expression of the pro-inflammatory tumor necrosis factor α (TNF-α) and interleukin-6 (IL-6). While the precise mechanism is unclear, the inhibition of miR-22 prevents the pathological upregulation of these two genes in hepatocytes and adipose tissue confirmed by in vitro and in vivo studies under HFD conditions [29,43].

The levels of miR-22 are inversely correlated with FGF21, and its receptor FGFR1 in human and mouse fatty livers, suggesting that hepatic miR-22 acts as a metabolic silencer [37,108]. The pharmacologic inhibition of miR-22 increases FGF21 and activates AMPK and PGC1α, reversing hepatic steatosis [37]. The same effect has been observed in a DIO-induced hepatic steatosis mouse model through the direct regulation of PPARα and SIRT1 by miR-22 [109]. Moreover, miR-22 expression has been demonstrated to be negatively correlated with FGF21 in adolescents with insulin resistance, where increased miR-22, blunting hepatic FGF21, promotes the worsening of steatosis [35]. These findings provide insights into the role of miR-22 in MASLD development, suggesting that miR-22 could be a potential therapeutic target for MASLD and obesity [43].

## 5. Potential Role of miR-22 in Myopathies

Beyond metabolic conditions, miR-22 is also implicated in muscle tissue pathology, specifically in muscular dystrophies (MDs). The overlapping molecular mechanisms shared between metabolic and muscular diseases underscore the broader regulatory role of miR-22 and open potential therapeutic applications within muscular dystrophies.

MDs are a group of genetic disorders that cause the progressive degeneration of the muscular system with widespread necrosis and the subsequent partial regeneration of the muscular tissue in response to an insult. The dystrophic cycle fosters inflammation and favors the accumulation of fibrotic and adipose tissue in the muscle fibers, progressively decreasing the overall contractile force. The pathophysiology of MDs therefore shares a similar molecular pattern with MASLD and other metabolic diseases. The most prevalent MDs are dystrophinopathies, characterized by a wide spectrum of loss-of-function mutations in the *DMD* gene that causes reduced to no synthesis of dystrophin, a protein crucial for the integrity of the sarcolemma. There is no cure for dystrophinopathies, and its most severe form, Duchenne muscular dystrophy (DMD), leads to premature death due to cardiac and respiratory failure.

Several miRNAs have been found to be associated with different MDs, mostly involved in myogenesis, muscle development, and regeneration [130,131,132,133]. In addition to their putative role as disease biomarkers, some have shown possible therapeutic potential, either by increasing the synthesis of the structural proteins, dystrophin and utrophin; being cardioprotective and anti-fibrotic; or by restoring calcium homeostasis [134]. Interestingly, miR-22 was reported to directly or indirectly control several targets involved in such processes, making it a possible therapeutic target for MDs. Primarily, miR-22 is involved in striated muscle repair and regeneration. It promotes myogenic differentiation in murine myoblasts following two signaling pathways; miR-22 directly inhibits histone deacetylase 4 (HDAC4), which is critical for skeletal muscle differentiation and regeneration through the de-repression of MEF2C [135], and targets transforming growth factor β receptor 1 (TGFβ-R1), therefore negatively modulating the TGF-β/SMAD pathway, a repressor of the MyoD family transcriptional factors [136]. Interestingly, mutual antagonism is exerted, in turn, by the profibrotic cytokine TGF-β1 on miR-22, preventing the cell proliferation function it has in myoblasts [136]. These results are in line with the progressive upregulation of miR-22 during myoblast differentiation and cardiomyocyte maturation [136,137]. Furthermore, miR-22 controls cardiac remodeling, inducing hypertrophy in cardiomyocytes. Although cardiac hypertrophy is physiological during the development [138], pathological hypertrophy driven by the dystrophic condition leads to increased thickness in the ventricular wall and an enlarged left ventricle, which result in cardiac dysfunction marked by a decreased left ventricle ejection fraction and arrhythmias [139]. Levels of miR-22 were reported to be elevated in cellular and mouse hypertrophic models by Zhan-Peng Huang et al., who also confirmed that the overexpression of miR-22 in cardiomyocytes is sufficient to develop hypertrophy and cardiomyopathy [137]. The enforced expression of miR-22 in cardiomyocytes results in reduced basal contractility, possibly negatively affecting calcium storage and release through the modulation of central regulators of calcium transients and fiber contraction [52]. Conversely, the hearts of mice congenitally defective for miR-22 show a significant reduction in hypertrophy when subjected to injury [137].

## 6. miR-22 as a Potential Biomarker in MASH

Besides its potential therapeutic role in the attenuation of adipogenesis and steatosis, miR-22 holds promise in the early identification of metabolic disorders such as MASLD and MASH. At present, it remains challenging to accurately identify the progression of the disease stage from MASLD, MASH, cirrhosis, and potentially to hepatocellular carcinoma (HCC) [140]. Clinically, the gold standard of diagnosis is still by liver biopsy [141], which is invasive for the patient and can lead to an inaccurate representation of the degree of lipid accumulation, inflammation, and fibrosis in the liver depending on the biopsy site [142]. Therefore, many indirect indications of disease are checked before taking a biopsy, such as measuring liver function and liver enzymes by blood sample or by imaging tests like ultrasound. A precise, non-invasive biomarker for MASLD/MASH progression could enable earlier detection and improved disease management.

miR-22 can be reliably quantified in human body fluids [143], which is important for accurately assessing miR-22 expression changes, and gene expression studies have identified both plasma and serum as useful sources of miRNAs [144], although serum should be preferred for clinical analysis as miRNAs are more stable over time [145]. Additionally, many current therapies for metabolic syndromes, including METF, ARBs, benzodiazepines, and OMEP, do not significantly affect miR-22 serum levels, yet miR-22 levels correlate with the severity of steatosis in MASLD patients [44,143]. This further supports the use for miR-22 as a biomarker in metabolic diseases like MASLD/MASH.

miR-22 is significantly overexpressed in the liver of human subjects with MASLD [29,35,37] and the accumulation of miR-22 in the blood is positively correlated with the steatosis grade in the liver [143]. Although these findings were made on drug-induced, iatrogenic steatosis, they imply that miR-22 could potentially be used to predict the progression of MASLD to MASH. Cook et al. showed that serum levels of miR-22 and miR-210 have prognostic predictive potential for hepatic fibrosis, suggesting that, combined, these miRNAs can accurately discriminate F1–F4 stages from F0 in humans [146]. A more specific sensitive prediction of late-stage fibrosis (F3–F4) was further achieved with a combination of miR-122, miR-21, miR-25, miR-210, miR-148a, and miR-19a, opening to a possible use of miRNAs as circulating biomarkers for the assessment of cirrhotic progression in MASLD. Importantly, serum miR-22 is present in two different forms: protein-bound and exosome-residing miR-22, respectively. Zhao et al. found that the number of EVs released from the liver increased dramatically during NAFLD under steatotic conditions [147], and several studies found a link between EV-secretion and various metabolic diseases [148]. Therefore, serum miR-22 should ideally be measured both as protein-bound (Ago2) and exosomal miRNA [7]. Using miR-22 in conjunction with other miRNAs as biomarkers would not only improve the early diagnosis of metabolic diseases with less invasive and highly precise tools but could also potentially predict the adverse effects of drug-induced steatosis [143].

## 7. miR-22 as a Therapeutic Target 

Building on the evidence of miR-22’s involvement across multiple metabolic and muscle-related pathways, its modulation has emerged as a promising strategy for therapeutic intervention in metabolic and muscular diseases.

When developing miRNA-targeted therapeutics, a commonly employed approach is to use antisense oligonucleotides (named antimiRs) to pharmacologically inhibit disease-implicated miRNAs [149]. Therapeutic antisense oligonucleotides (ASOs) include different chemical modifications to improve target affinity and stability while reducing toxicity [150]. Modifications of the phosphodiester bond—especially phosphorothioate (PS) linkages—and different sugar modifications, such as 2′-O-methyl (2′-O-Me), 2′-methoxyethyl (2′-MOE), cyclic ethyl (cEt) or locked nucleic acid (LNA), are deployed to improve resistance to nucleases, enhance pharmacokinetic properties, and increase binding affinity of the ASOs to the cognate RNA target [149,151,152]. So-called mixmer ASOs, in which high affinity nucleotides such as LNAs and unmodified DNA nucleotides are interspersed across the ASO sequence, have become the choice in antimiR oligonucleotides [153,154], as they allow for the effective steric blockage of miRNAs, thereby inhibiting their biological function [155].

Several antimiR-based drugs have already entered clinical trials, including miravirsen, cobomarsen, and CDR132L, which are all LNA-modified mixmer ASOs [154,156,157]. Miravirsen (SPC3649) was the first antimiR drug to enter clinical trials and represented a new paradigm in the treatment of hepatitis C virus by targeting miR-122, a host factor important for the stability of the viral genome, its replication, and translation [158]. Extensive testing in phase II trials showed the significant and sustained dose-dependent inhibition of the viral load in patients, along with some mild or moderate side effects [159]. Cobomarsen is an antimiR designed to inhibit miR-155. The compound has been extensively tested for safety and efficacy, undergoing one phase I and two phase II trials for the treatment of both cutaneous T-cell lymphoma/mycosis fungoides and adult T-cell leukemia (NCT02580552, NCT03713320, NCT03837457) [160]. Although both phase II trials were prematurely terminated, no serious adverse events were reported, and the termination of the program was a business decision [157]. CDR132L, which is an antimiR developed by Cardior to target miR-132, is intended to halt and reverse the detrimental cardiac remodeling that follows myocardial infarction (MI). Being currently investigated in a phase 2 clinical trial (NCT05350969), Cardior’s antimiR was found to be safe and well-tolerated in humans. Treatment with the CDR132L compound has also resulted in improved cardiac function in patients with heart failure by shortening the pathologically prolonged action potential [161].

The abovementioned phase 2 trials and other antimiRs in clinical testing are encouraging for similar therapeutic approaches to become viable for the treatment and/or to halt the progression of MASLD to MASH.

Hu et al. demonstrated that the inhibition of miR-22 in healthy mice (1 × 10^9^ PFU tail-vein injection once a week for 4 months) led to no difference in body weight gain or liver-to-body weight ratio in both male and female mice, and that there were no altered serum levels of ALT, ALP or LPS [37]. Inhibiting miR-22, however, did reduce fasting blood glucose levels in male mice, suggesting a metabolic benefit even under healthy conditions. This is consistent with other studies that compared male and female metabolism; it seems that healthy human and rodent females are better protected from metabolic diseases than healthy males, likely due to estrogen having a positive effect [162,163]. Therefore, a visible change in blood glucose levels could be easier to induce in male mice.

A study by Xu, Liu, and Zhang showed that the pharmacological inhibition of miR-22 prevented oxidative stress via upregulating SIRT1 [164], and although this study was conducted in the heart, other studies point towards neutrophil infiltration of the liver, accompanied by oxidative stress, to be one of the drivers of MASLD to MASH progression [165]. Thus, raising the levels of SIRT1 via miR-22 inhibition could be attractive in the context of the liver as well.

Several miRNAs have shown possible therapeutic potential in dystrophinopaties, either by increasing the synthesis of structural proteins, dystrophin and utrophin; being cardioprotective and anti-fibrotic; or restoring calcium homeostasis [134]. miR-22 was observed directly and indirectly controlling several genetic targets involved in such processes, making it a possible therapeutic target for MDs. Pgc-1a overexpression has a protective effect against muscle dystrophy, dramatically improving muscle damage in *mdx* mice even when completely lacking utrophin (mdx/utrn^−/−^), regardless of whether the overexpression is induced pre- or postnatally [166]. miR-22 is a well-known inhibitor of PGC-1a through SIRT-1 and the inhibition of miR-22 has been characterized to increase the synthesis of PGC-1a in different tissues, including striated muscles [52,94]. Furthermore, miR-22 was observed to be a positive regulator of hypertrophy in the heart following cardiac insult [138,139]. Interestingly, hindering hypertrophy via miR-22 loss showed opposite effects in different mouse models of cardiac injury. Congenital ablation of miR-22 was reported to negatively influence the recovery of cardiac insult, increasing the susceptibility to stress-mediated dilated cardiomyopathy as well as cell apoptosis in adult mice [137]. On the other hand, the pharmacologic inhibition of miR-22 following myocardial infarction in old mice was proven to be beneficial in restoring cardiac function, reversing ejection fraction, cardiac output, and systolic volume to pre-insult levels [167,168]. Therefore, the potential therapeutic effect of miR-22 inhibition for the treatment of cardiomyopathy might depend on multiple factors, among which are those that regulate the increment of miR-22 in cardiomyocytes, such as age, as well as therapeutic aspects like the induction timing and the extension of the miRNA inhibition. Moreover, contrary to the aforementioned models, MDs are a source of continuous cardiac insult; therefore, more studies are needed to elucidate the possible effect of miR-22 inhibition in these conditions.

## 8. Conclusions and Future Prospects

In this review, we discussed the role of miR-22 in metabolic diseases such as MASH, obesity, and diabetes, implicating its function in altering gluconeogenesis, thermogenesis, the browning of adipocytes, insulin sensitivity, glucose homeostasis, fatty acid biosynthesis, and lipid catabolism. Present data are consistent with the notion of miR-22 being a master regulator of lipogenesis, and thereby highlight its potential as a therapeutic target for the treatment of steatosis and insulin resistance (see [37]). Furthermore, several lines of evidence suggest that miR-22 could be a useful biomarker for metabolic diseases. Firstly, it is highly expressed in all the major organs and tissues responsible for energy metabolism. Secondly, the levels of miR-22 were shown to be influenced by metabolic dysfunction. Thirdly, miR-22 levels are able to stratify MASH patients for early and late-stage liver fibrosis, respectively. Finally, the possibility to measure miR-22 directly in serum samples, where its levels reflect disease-associated patterns in tissues, underscores its potential as a robust, non-invasive biomarker for metabolic diseases.

Several studies have reported that miR-22 is overexpressed in mice and human subjects with obesity, MAFLD, and diabetes, and that its inhibition in mice leads to significant improvements in outcome, as measured by biomarkers related to the abovementioned diseases. In addition, increasing evidence implies that these diseases are often correlated [109]: more than 90% of the severely obese subjects (bariatric surgery patients) also have MASLD [108]. This effect has been confirmed by analyses of the interactions between miR-22 and multiple downstream pathways, underscoring the beneficial effect of reducing miR-22 expression levels to improve metabolism. A growing body of literature has shown the importance of miR-22 as a key regulator of systemic glucose and hepatic lipid metabolism. Thus, miR-22 inhibition may represent a novel therapeutic strategy for the treatment of metabolic diseases. Indeed, several studies have implied miR-22 inhibitors as a novel therapy to control obesity, type 2 diabetes mellitus, and MASLD [29,35,37,39,40,41,42,44,74,108], which is highlighted by the beneficial effects of miR-22 inhibition in reverting the pathological dysregulation of gluconeogenesis, lipogenesis, and mitochondrial biogenesis. While the therapeutic modulation of miR-22 shows great promise, several challenges remain before translation into clinical practice. These include developing tissue-specific delivery systems to ensure effective targeting, minimizing potential off-target effects, and maintaining the stability and efficacy of miR-22-based therapies in vivo. Addressing these challenges will require robust preclinical validation, alongside well-designed early-phase clinical trials, to fully realize the therapeutic potential of miR-22. Future research focusing on the translational potential of miR-22 inhibition, the targeted delivery of miR-22 inhibitors with enhanced specificity, and testing of different combination strategies with current therapies will be essential for advancing miR-22-targeted therapeutics from metabolically relevant animal models to clinical development.

## 9. Methods

This work is based on the consulted literature from ScienceOpen, Science Direct, Scopus, and Google Scholar during 2024, based on the following keywords: miR-22, microRNA-22, microRNA, miRNA, liver, muscle, heart, adipose, metabolism, metabolic. Results were filtered to focus on articles published between 2010 and 2024. Older articles and reviews were only considered regarding established knowledge not directly related to miR-22. Further filtering of the material was performed to focus on metabolically relevant diseases, avoiding cancer-related material.

## Figures and Tables

**Figure 1 ijms-26-00782-f001:**
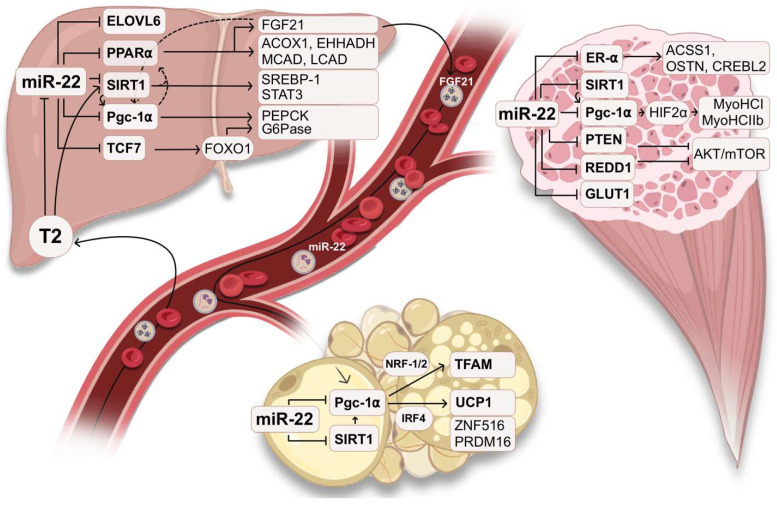
The physiological role of miR-22 in metabolism. miR-22 is a metabolic regulator that mediates the transcriptional silencing of key genes involved in gluconeogenesis, lipogenesis, fatty acid oxidation (FAO), and the beiging of white adipose tissue (WAT). Hepatic glucose production is induced in the liver under the influence of thyroid hormones such as 3,5-diiodo-L-thyronine (T2), possibly through the direct activation of SIRT1 and inhibition of miR-22, further inducing the expression of PEPCK and G6Pase via PGC-1α and TCF7 increases. miR-22 also inhibits the peroxisomal and mitochondrial β oxidation directly targeting PPAR-α, which induces the secretion of FGF21 to control the expression of *PPARGC1A* in a paracrine and endocrine manner, respectively, in WAT. Here, the activation of PGC-1α through SIRT1 modulates TFAM and UCP1, increasing mitochondrial biogenesis and energy expenditure, respectively, cell programs characteristic of the more metabolically active brown adipose tissue. In skeletal muscle, miR-22 is responsible for the inhibition of lipid catabolism via estrogen receptor alpha (ER- α) and for promoting anabolic pathways via mTOR derepression. miR-22 also controls muscle fiber switch, favoring fast-twitching glycolytic fibers, and it can be released in the bloodstream within extracellular vesicles (EVs) to post-transcriptionally control the expression of its targets in distal organs. This figure was based on BioRender templates.

**Table 1 ijms-26-00782-t001:** List of the principal miR-22 targets involved in the metabolic rewiring mediated by the inhibition of miR-22 and their role in the modulation of energy metabolism in different tissues.

miR-22 Target	Target Validation	Tissue	Pathway	Enzymes Responsible for Pathway Regulation	Effect of miR-22 Inhibition on Metabolism	References
** *PPARGC1A* **	[37,52]	Liver	Gluconeogenesis	PEPCK, G6Pase	miR-22 deletion reverses the inhibitory effect of FFA/CDAA/western diet on gluconeogenesis by increasing the expression of gluconeogenic enzymes in vitro and in vivo via PGC-1α.	[37,48]
		Adipose	Non-shivering thermogenesis	UCP-1	Deletion of miR-22 protects mice from impairments of mitochondrial respiration induced in BAT by HFD preventing UCP-1 reduction and stimulating the thermogenic program and energy expenditure.	[40]
		Muscle	Mitochondrial biogenesis	TFAM	miR-22 deletion restores the expression of several genetic markers of mitochondrial biogenesis to pre-obesogenic levels	[29]
			Fiber type conversion	MEF2	miR-22 inhibition induces conversion of fast-twitch muscle fibers to slow-twitch fibers by boosting mitochondrial oxidative metabolism.	[94,95]
** *SIRT1* **	[43,52]	Liver	Gluconeogenesis	PGC-1α	miR-22 inhibition and deletion results in increased expression of SIRT1 in different mouse models, activating PGC-1α.	
		Adipose	Mitochondrial biogenesis	PGC-1α		
		Muscle	Mitochondrial biogenesis	PGC-1α		
** *TCF7* **	[39]	Liver	Gluconeogenesis	PEPCK, G6Pase	miR-22 inhibition decreases glucose production via direct derepression of TCF7, lowering the dysregulated fasting circulatory glucose levels of diabetic *db*/*db* mice.	[39,96]
** *ELOVL6* **	[64]	Adipose	Thermogenesis	TfR1	miR-22 deletion counteracts the impairment in WAT browning exerted by HFD, inducing higher expression of BAT markers among which *Elovl6*, necessary for appropriate mitochondrial function.	[40,97]
** *PPARA* **	[52]	Liver	mitochondrial fatty acid β-oxidation	SIRT1	miR-22 inhibition induces a general increase in expression of *Ppara* in conjunction with genes involved in hepatic lipid catabolism (*Acadm*, *Cpt1a*), and fatty acid synthesis (*SREBP1*) and oxidation (*FOXO1*).	[43,98]
** *ESR1* **	[77,99]	Muscle	Lipid metabolism		miR-22 hinders weight increase during development in male mice only, reducing WAT accumulation in skeletal muscles through ERα repression.	[77]
** *FGFR1* **	[37]	Liver	Lipid metabolism	FGF21	miR-22 inhibition improves FGF21 signalling alone and combined with FGF21 inducers to reduce hepatic steatosis in western diet-fed mice.	[37]

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
