# Peer review of "The Role of microRNA-22 in Metabolism"

_ijms, 2025, doi:10.3390/ijms26020782_

Round 1

Reviewer 1 Report

Comments and Suggestions for Authors

Comments to the Authors of manuscript number ijms-3370058  entitled “The role of microRNA-22 in metabolism”

1. The abstract claims that miR-22 modulation holds significant potential for translational research and therapeutic applications in metabolic disorders. However, it later suggests that miR-22 can have conflicting roles (e.g., promoting or inhibiting processes depending on context). This dual role is not acknowledged or clarified in the abstract, making it overly optimistic and lacking nuance.

2. "transcriptional regulation" is inaccurate

3. The introduction notes that both strands of the miR-22 duplex are released, but miR-22-3p predominates. The text does not explain why miR-22-5p is less significant, which is critical for understanding its relative contributions.

4. The contradiction in the role of miR-22 is not reconciled or addressed clearly, making the role of miR-22 seem inconsistent.

5. The text states that miR-22 drives fiber-type conversion in muscle by negatively regulating the AMPK/SIRT1/Pgc-1α axis. However, it also claims that inhibition of miR-22 promotes oxidative fibers, which is linked to improved aerobic capacity. This directly conflicts with the earlier assertion and creates confusion about whether miR-22 has a positive or negative role in promoting aerobic muscle metabolism.

6. Many sections claim therapeutic potential of miR-22 modulation (e.g., in MASLD, diabetes, obesity, and MDs) without presenting clinical evidence or discussing challenges in therapeutic targeting

7. Lack of Clarity on SREBP-1c Regulation

8. The text oversimplifies the interplay of adipogenesis-related genes, such as PPARG and SREBP-1c. It fails to provide a cohesive mechanism of how miR-22 distinctly influences these pathways under different conditions.

9. "Inhibition of miR-22 can produce fat mass reduction without reducing food intake or acting on body temperature." It incorrectly suggests that miR-22 itself accumulates in the blood as a result of liver damage.

10. miR-22 inhibition affects hypertrophy differently in dystrophic conditions – it should be clearer presented

11. "miR-22 is a positive regulator of hypertrophy in the heart" this regulation is context-dependent and differs in pathological versus normal conditions

12. the part of material and methods is needed describing what keywords and databases were used. The description of searching also is expected.

Author Response

Response to reviews of the manuscript ijms-3370058 by Tomasini et al. entitled entitled “The role of microRNA-22 in metabolism”.

We would like to thank the reviewers for their objective and thorough review of our manuscript. Please, find below our detailed response to the reviews. We sincerely believe that we have adequately ad-dressed all the comments raised by the reviewers and, thus, hope that the revised manuscript in its present form could be accepted for publication in International Journal of Molecular Sciences

Comments to the Authors of manuscript number ijms-3370058  entitled “The role of microRNA-22 in metabolism”

Comment 1: The abstract claims that miR-22 modulation holds significant potential for translational research and therapeutic applications in metabolic disorders. However, it later suggests that miR-22 can have conflicting roles (e.g., promoting or inhibiting processes depending on context). This dual role is not acknowledged or clarified in the abstract, making it overly optimistic and lacking nuance.

Response 1: Thank you for rising this point. In the revised manuscript, we have changed the abstract to account for the nuances present in the body of the review article The revised abstract follows:

“MicroRNA-22 (miR-22) plays a pivotal role in the regulation of metabolic processes and has emerged as a therapeutic target in metabolic disorders, including obesity, type 2 diabetes, and metabolic-associated liver diseases. While miR-22 exhibits context-dependent effects, promoting or inhibiting metabolic pathways depending on tissue and condition, current research highlights its therapeutic potential, particularly through inhibition strategies using chemically modified antisense oligonucleotides. This review examines the dual regulatory functions of miR-22 across key metabolic pathways, , offering perspectives on its integration into next-generation diagnostic and therapeutic approaches while acknowledging the complexities of its roles in metabolic homeostasis.”

Comment 2: "transcriptional regulation" is inaccurate

Response 2: We agree. We have, accordingly, modified "transcriptional regulation" with “the silencing function” (line 32).

Comment 3: The introduction notes that both strands of the miR-22 duplex are released, but miR-22-3p predominates. The text does not explain why miR-22-5p is less significant, which is critical for understanding its relative contributions.

Response 3: We revised lines 40-48 to emphasize the biological reason underlying the predominance of the 3p strand over the 5p.

Comment 4: The contradiction in the role of miR-22 is not reconciled or addressed clearly, making the role of miR-22 seem inconsistent.

Response 4: Thank you for pointing out the confusing content in the introduction. We modified the conclusive lines 70-73 to better reconcile the regulatory activity of miR-22 acting on different metabolic-related pathways and tissues, and its therapeutic potential. The updated text is reported below:

“Notably, miR-22 exhibits a context-dependent regulatory role, influencing metabolic and muscular pathways in distinct and tissue-specific manners. Understanding how miR-22 integrates these signals is critical to fully elucidating its therapeutic potential.”

Furthermore, we added a new table to better present the metabolic effects induced by miR-22 inhibition, singularly clustered according to primary target, tissue of origin, and signaling pathway involved.

Comment 5: The text states that miR-22 drives fiber-type conversion in muscle by negatively regulating the AMPK/SIRT1/Pgc-1α axis. However, it also claims that inhibition of miR-22 promotes oxidative fibers, which is linked to improved aerobic capacity. This directly conflicts with the earlier assertion and creates confusion about whether miR-22 has a positive or negative role in promoting aerobic muscle metabolism.

Response 5: Thank you for your comment. We modified lines 149-157 accordingly to better clarify the physiological role of miR-22 in fiber-type conversion and the therapeutic effect of miR-22 inhibition. The updated paragraph follows:

“miR-22 regulates muscle fiber composition by modulating the AMPK/SIRT1/PGC-1⍺ axis. High levels of miR-22 promote the enrichment of fast-twitch, type II glycolytic myofibers, which are associated with reduced oxidative capacity. Conversely, inhibition of miR-22 leads to an increase in slow-twitch oxidative fibers (type I), characterized by enhanced mitochondrial biogenesis and improved aerobic capacity. Improvement of muscle aerobic capacity has been shown to be dependent on SIRT1/PGC-1⍺ activation, with increased expression of type I fiber-specific genes and mitochondrial tissue density upon treatment with PGC-1⍺ activators such as resveratrol75,76

Comment 6: Many sections claim therapeutic potential of miR-22 modulation (e.g., in MASLD, diabetes, obesity, and MDs) without presenting clinical evidence or discussing challenges in therapeutic targeting.

Response 6: Thank you for rising this concern. Although we strongly believe in the therapeutic potential of targeting miR-22, we agree this therapeutic strategy presents limitations and challenges still to be overcome. Therefore, we improved the text by inserting lines 558-564 – reported below – to improve clarity and provide suggestions to address our main concerns.

“While the therapeutic modulation of miR-22 shows great promise, several challenges remain before translation into clinical practice. These include developing tissue-specific delivery systems to ensure effective targeting, minimizing potential off-target effects, and maintaining stability and efficacy of miR-22-based therapies in vivo. Addressing these challenges will require robust preclinical validation, alongside well-designed early-phase clinical trials, to fully realize the therapeutic potential of miR-22”

Comment 7/8: Lack of Clarity on SREBP-1c Regulation/The text oversimplifies the interplay of adipogenesis-related genes, such as PPARG and SREBP-1c. It fails to provide a cohesive mechanism of how miR-22 distinctly influences these pathways under different conditions.

Response 7/8: We agree with you. Therefore, we revised and improved the three paragraphs (pages 5-6, lines 250-282) related the control mechanism exerted by miR-22 on SREBP-1c and PPARG to better convey its role in adipogenesis.

Comment 9: "Inhibition of miR-22 can produce fat mass reduction without reducing food intake or acting on body temperature." It incorrectly suggests that miR-22 itself accumulates in the blood as a result of liver damage.

Response 9: Thank you for pointing out the incorrect underlying message. We corrected the conclusive paragraph (lines 310-315) relative to the “Role of miR-22 in diabetes and obesity” stating it more clearly as follows:

“Taken together, these findings suggest that inhibition of miR-22 function can reduce fat mass without affecting food intake or body temperature.41 Importantly, there is no evidence linking this effect to the accumulation of miR-22 in the bloodstream as a consequence of liver damage. Instead, it reflects the role of miR-22 in the regulation of adipogenesis and energy homeostasis.”

Comment 10: miR-22 inhibition affects hypertrophy differently in dystrophic conditions – it should be clearer presented

Response 10: Thank you for your suggestion. The literature related to the role of miR-22 in muscular dystrophy is scarce. For this reason, we focused on its potential role based on works describing miR-22 function in biological processes linked to the pathophysiology of muscular dystrophy, and particularly DMD. We modified the last part of the chapter (lines 401-413) to better clarify the use of models of cardiac hypertrophy and not muscular dystrophy.

Further complexity in the effects mediated by miR-22 inhibition on cardiac hypertrophy were added (lines 517-527) and attention was drowned on the potential drawbacks in using models of cardiac hypertrophy and myocardial infarction for modelling of muscular dystrophy (lines 527-530).

Comment 11: "miR-22 is a positive regulator of hypertrophy in the heart" this regulation is context-dependent and differs in pathological versus normal conditions

Response 11: We agree that the previous message was misleading. We have therefore specified that “miR-22 was observed being a positive regulator of hypertrophy in the heart following cardiac insult”(lines 516-517).

Comment 12: the part of material and methods is needed describing what keywords and databases were used. The description of searching also is expected.

Response 12: We added a new Methods section (lines 570-576).

Additional clarifications

The new manuscript also contains keywords for improved indexing

Reviewer 2 Report

Comments and Suggestions for Authors

Dear authors,

Your review 'The role of microRNA-122 in metabolism' presents a consolidated version of updated literature information regarding the participation of microRNA (miR) 122 in several metabolism processes including obesity, diabetes, metabolic dysfunction-associated steatotic liver disease (MALD), and steatohepatitis (MASH). I believe this review could be insightful to their field. Nevertheless, I would like to comment on some concerns and suggestions.

Major comments

1. Research in miRNAs and their targets is growing, and target information could be updated. You described approx 3000 miR-122 targets. Nevertheless, it is important to discuss validated targets, especially in experiments performed in metabolism-related cells.

2. miR-22 is highly conserved across species. However, the proteins and genes affected should be formatted according to the HUGO guidelines (https://www.genenames.org/about/guidelines/). For non-human species, genes and proteins should show only the first letter capitalized. Be sure that correct formatting was applied in all mentions.

3.  Some descriptions are related to the effect of miR-22 in the occurrence of metabolic disturbs. But others are effects on miR-22 levels produced by potential treatments. For example, "Administration of T2 activates hepatic SIRT1 directly and causes a decrease in miR-22 levels in both serum and liver of rats fed on a diet with high fat content". Please describe separately the causes or effects of miR-22 in the metabolic context.

4. The section on therapeutic proposals involving miR-22 should address administration strategies to improve the biodistribution of antimiRs until the corresponding cell. How should we translate in vivo experiments (for example, those with antimiR injected in the tail of mice) to a clinical setup? Would the use of nanolipoparticles or extracellular vesicles be recommended?

Minor comments

5. Please delete additional points in "which is highlighted .. by the beneficial effects of miR-22 inhibition in reverting pathological dysregulation of gluconeogenesis"

Author Response

Response to reviews of the manuscript ijms-3370058 by Tomasini et al. entitled entitled “The role of microRNA-22 in metabolism”.

We would like to thank the reviewers for their objective and thorough review of our manuscript. Please, find below our detailed response to the reviews. We sincerely believe that we have adequately ad-dressed all the comments raised by the reviewers and, thus, hope that the revised manuscript in its present form could be accepted for publication in International Journal of Molecular Sciences

Comments to the Authors of manuscript number ijms-3370058  entitled “The role of microRNA-22 in metabolism”

Comments 1: Research in miRNAs and their targets is growing, and target information could be updated. You described approx 3000 miR-122 targets. Nevertheless, it is important to discuss validated targets, especially in experiments performed in metabolism-related cells.

Response 1: Thank you for your suggestion. We would like to clarify that this review is related to the role of miR-22, a different microRNA than miR-122. We agree with you that we should focus on validated targets of miR-22, and we believe our work mainly describes physiological and disease-dysregulated processes relying on metabolic-related pathways controlled by direct targets of miR-22. Therefore, we modified the sentence (lines 80-83) as follows to delete the misleading information regarding the number of predicted miR-22 targets:

“Gene Ontology (GO) analysis of predicted targets of miR-22 conserved in humans and mouse uncovers a strong metabolic and regenerative component with a high enrichment in the central carbon metabolism pathway”

Furthermore, we added a new table to better present the metabolic effects induced by miR-22 inhibition, singularly clustered according to primary target, tissue of origin, and signaling pathway involved.

Comments 2: miR-22 is highly conserved across species. However, the proteins and genes affected should be formatted according to the HUGO guidelines (https://www.genenames.org/about/guidelines/). For non-human species, genes and proteins should show only the first letter capitalized. Be sure that correct formatting was applied in all mentions

Response 2: Agree. We have, accordingly, modified the text to format genes and protein names according to the relevant species following HGNC and MGNC guidelines. MGNC guidelines suggests gene symbols to be italicized beginning with an upper-case letter, and for protein symbols to use all uppercase not italicized letters.

Comments 3: Some descriptions are related to the effect of miR-22 in the occurrence of metabolic disturbs. But others are effects on miR-22 levels produced by potential treatments. For example, "Administration of T2 activates hepatic SIRT1 directly and causes a decrease in miR-22 levels in both serum and liver of rats fed on a diet with high fat content". Please describe separately the causes or effects of miR-22 in the metabolic context.

Response 3: Thank you for rising this point. We have now moved the description of the effects on metabolic pathways mediated by treatments targeting miR-22 in a separate paragraph within the “miR-22 as a regulator of key metabolic factors” chapter (lines 181-202).

Comments 4: The section on therapeutic proposals involving miR-22 should address administration strategies to improve the biodistribution of antimiRs until the corresponding cell. How should we translate in vivo experiments (for example, those with antimiR injected in the tail of mice) to a clinical setup? Would the use of nanolipoparticles or extracellular vesicles be recommended?

Response 4: Thank you for rising this concern. Administration strategies for antimiRs were not discussed as we deemed it out of the scope of this work. We inserted lines 558-564 – reported below – to improve clarity and provide suggestions to address translatability into clinic of the research presented.

“While the therapeutic modulation of miR-22 shows great promise, several challenges remain before translation into clinical practice. These include developing tissue-specific delivery systems to ensure effective targeting, minimizing potential off-target effects, and maintaining stability and efficacy of miR-22-based therapies in vivo. Addressing these challenges will require robust preclinical validation, alongside well-designed early-phase clinical trials, to fully realize the therapeutic potential of miR-22”

Comments 5: Please delete additional points in "which is highlighted .. by the beneficial effects of miR-22 inhibition in reverting pathological dysregulation of gluconeogenesis"

Response 5: Thank you, we have corrected the mistake (lines 556-557).

Additional clarifications

The new manuscript also contains keywords for improved indexing.

Round 2

Reviewer 2 Report

Comments and Suggestions for Authors

Dear authors,

Your review 'The role of microRNA-22 in metabolism' presents a consolidated version of updated literature information regarding the participation of microRNA (miR)-22 in several metabolism processes including obesity, diabetes, metabolic dysfunction-associated steatotic liver disease (MALD), and steatohepatitis (MASH). I believe this review could be insightful to their field. Thank you for having addressed my previous concerns.